# Current Smoking Determines the Levels of Circulating MPO and MMP-9 in Adults with Coronary Artery Disease and Obstructive Sleep Apnea

**DOI:** 10.3390/jcm12124053

**Published:** 2023-06-14

**Authors:** Esra Özkan, Yeliz Celik, Tülay Yucel-Lindberg, Yüksel Peker

**Affiliations:** 1Graduate School of Health Sciences, Koc University, Istanbul 34450, Turkey; eozkan19@ku.edu.tr (E.Ö.); yecelik@ku.edu.tr (Y.C.); 2Research Center for Translational Medicine [KUTTAM], School of Medicine, Koc University, Istanbul 34450, Turkey; 3Columbia University Irving Medical Center, New York, NY 10032, USA; 4Division of Pediatric Dentistry, Department of Dental Medicine, Karolinska Institute, 141 86 Huddinge, Sweden; tulay.lindberg@ki.se; 5Department of Molecular and Clinical Medicine, Institute of Medicine, Sahlgrenska Academy, University of Gothenburg, 405 30 Göteborg, Sweden; 6Division of Sleep and Circadian Disorders, Brigham and Women’s Hospital & Harvard Medical School, Boston, MA 02115, USA; 7Division of Pulmonary, Allergy, and Critical Care Medicine, School of Medicine, University of Pittsburgh, Pittsburgh, PA 15213, USA; 8Department of Clinical Sciences, Respiratory Medicine and Allergology, Faculty of Medicine, Lund University, 220 02 Lund, Sweden

**Keywords:** coronary artery disease, obstructive sleep apnea, myeloperoxidase, matrix metalloproteinase 9, current smoking

## Abstract

(1) Background: Obstructive sleep apnea (OSA) is common in patients with coronary artery disease (CAD), in which a rupture of atherosclerotic plaques and oxidative stress play a role in the initiation and progression of the disorder. Circulating levels of myeloperoxidase (MPO), as an oxidative stress marker, as well as matrix metalloproteinase-9 (MMP-9), as a destabilizer of plaques, are known to be elevated in patients with CAD and are associated with worse prognosis. Some studies have suggested that OSA is associated with MPO and MMP-9, but the effect of OSA on these biomarkers in cardiac cohorts is unknown. (2) Aims: We addressed the determinants of high MPO and MMP-9 in a CAD cohort with concomitant OSA. (3) Materials and Methods: The current study was a secondary analysis of the RICCADSA trial that was conducted in Sweden between 2005 and 2013. A total of 502 revascularized CAD patients with OSA (apnea–hypopnea index [AHI] ≥ 15 events/h; *n* = 391) or no-OSA (AHI < 5 events/h; *n* = 101), based on a home sleep apnea test, and who had blood samples at baseline were included in the analysis. The patients were dichotomized into a high or low MPO and MMP-9 groups, based on the median cut-off values. (4) Results: The mean age of the participants was 63.9 (±8.6), and 84% of the study cohort were men. Median values of MPO and MMP-9 levels were 116 ng/mL and 269 ng/mL, respectively. In different multivariate linear and logistic regression models, neither OSA nor OSA severity in terms of AHI and oxygenation indices were associated with the high MPO and MMP-9 levels. Current smoking was significantly associated with both high MPO (odds ratio [OR] 1.73, 95% confidence interval [CI] 1.06–2.84; *p* = 0.030) and high MMP-9 levels (OR 2.41, 95% CI 1.44–4.03; *p* < 0.001), respectively. Other significant determinants were revealed as beta blocker use (OR 1.81, 95% CI 1.04–3.16; *p* = 0.036) for high MPO as well as male sex (OR 2.07, 95% CI 1.23–3.50; *p* = 0.006) and calcium antagonist use (OR 1.91, 95% CI 1.18–3.09; *p* = 0.008) for high MMP-9 levels. (5) Conclusions: Current smoking, but not OSA, was significantly associated with high MPO and MMP-9 levels in this revascularized CAD cohort. Smoking status should be seriously taken into consideration while evaluating the effects of OSA and its treatment on long-term adverse cardiovascular outcomes in adults with CAD.

## 1. Introduction

Obstructive sleep apnea (OSA), which is characterized by recurrent episodes of an upper airway collapse leading to partial or total loss of breathing, intermittent hypoxemia, and arousals, affects 34% of middle-aged men and 17% of middle-aged women in the general population [1] and almost 63% of the adults with coronary artery disease (CAD) [2]. OSA was also shown to be related to cardiovascular risk factors [3], arrhythmias [4], reduced hearth rate variability [5], as well as structural cardiac phenotypes associated with higher CAD risk such as right ventricular strain and coronary artery ectasia [6,7]. Most importantly, OSA may accelerate the cardiovascular disease’s progression, contributing to the worse outcomes and mortality [8]. Rupture of atherosclerotic plaques, endothelial dysfunction, and oxidative stress are known to play a role in the initiation and progression of CAD [9]. In this context, circulating levels of myeloperoxidase (MPO) and matrix metalloproteinase-9 (MMP-9) have a particular importance [9].

MPO is an enzyme located in azurophilic granules of neutrophils and monocytes [10]. MPO contains a heme-group, works as a peroxidase in inflammatory pathways, and triggers oxidative stress [10]. Studies have demonstrated that MPO levels are higher in CAD [11,12], and elevated circulating levels of MPO in acute coronary syndrome have been shown to be associated with a worse prognosis in those patients [13]. Moreover, it has been suggested that increased MPO levels predict future CAD events [14]. 

MMP-9 is known to be a zinc containing endopeptidase, responsible for degrading the extracellular matrix and essential in cardiac and vascular remodeling [15]. In addition, MMP-9 is involved in the instability and rupture of atherosclerotic plaques [16]. Circulating levels of MMP-9 are also known to be elevated in patients with CAD [17,18,19] and can predict mortality in those individuals [20].

Studies regarding the relationship between OSA and MPO are controversial since some suggest a significant association [21,22] whereas some others do not [23]. However, most of the studies addressing the relationship between OSA and MMP-9 have been positive [24,25,26,27]. Not only the occurrence but also the severity of OSA has been suggested to be associated with the increasing levels of the circulating MMP-9 levels in sleep clinic cohorts [24,26,27], as also reported in a meta-analysis [28].

Less is known regarding the effect of OSA on these biomarkers in cardiac cohorts. Circulating MMP-9 levels were found to be elevated among patients with CAD and concomitant OSA in a small Turkish case–control study [29]. Recently, no significant differences were found regarding the circulating MPO and MMP-9 levels in 209 CAD patients with OSA compared to the levels in 152 patients without OSA among the participants of the ISAACC (CPAP in Patients with Acute Coronary Syndrome and OSA) trial [30]. Of note, almost half of the ISAACC study cohort were current smokers at baseline, which might have contributed to neutral findings with regard to the composite endpoints in that study.

The RICCADSA (Randomized Intervention with Continuous Positive Airway Pressure [CPAP] in CAD and OSA) trial primarily addressed the effects of CPAP on the composite of repeat revascularization, myocardial infarction, stroke, and cardiovascular mortality in revascularized patients with CAD and non-sleepy OSA [31,32,33]. The proportion of current smokers at baseline were 17% among the patients with OSA and 26% in no-OSA in the RICCADSA cohort, which makes the current work important to address the determinants of high circulating levels of MPO and MMP-9 in a larger sample of CAD patients with OSA versus without OSA, which has been the main aim of the current study.

## 2. Materials and Methods

### 2.1. Study Participants

This study is a secondary analysis of the RICCADSA trial. The methodology of the main trial has been explained in detail previously [31,32,33]. In brief, CAD patients who underwent revascularization (percutaneous coronary intervention or CABG) in Skaraborg County, West Sweden, were invited to participate [30]. All participants were investigated with home sleep apnea testing (HSAT) in a stable condition following the revascularization procedure. The patients were recruited between December 2005 and November 2010, and the final follow-up was in May 2013. As previously described in detail [31,32,33], inclusion criteria comprised of revascularized CAD patients with OSA (apnea–hypopnea index [AHI] of at least 15/h or no-OSA (AHI < 5/h)). Patients with borderline OSA (AHI ≥ 5 and <15/h) and the ones with dominantly Cheyne–Stokes respiration constituted the exclusion criteria. As illustrated in Figure 1, 244 CAD patients with OSA were randomized to CPAP or no-CPAP if they did not have excessive daytime sleepiness (Epworth Sleepiness Scale [ESS] score < 10) whereas 155 OSA patients with ESS score ≥ 10 who were offered CPAP and 112 patients without OSA were included in the observational arm (Figure 1). In all, 502 participants with plasma MPO and MMP-9 levels at baseline were eligible for the current study.

### 2.2. Home Sleep Apnea Test

For sleep studies, the Embletta^®^ Portable Digital System device (Embla, Broomfield, CO, USA) was used as explained previously [31]. The HSAT system consists of a nasal pressure detector and two respiratory inductance plethysmography belts for recording thoracoabdominal movements and body position. Additionally, heart rate and oxyhemoglobin saturation (SpO2) are recorded with a finger pulse oximeter. Cessation of airflow of at least 90% was accepted as apnea, and hypopneas as a >50% reduction in thoracoabdominal movement and/or a >50% decrease in the nasal pressure amplitude for >10 s regardless of oxygen desaturation (Chicago criteria) [34]. The oxygen desaturation index (ODI) was calculated as the number of significant drops in SpO2 exceeding 4% from the immediately preceding baseline per hour.

### 2.3. Epworth Sleepiness Scale

Patients filled out the ESS questionnaire [35] to assess subjective daytime sleepiness. The ESS consists of eight items questioning the risk of falling asleep under eight different circumstances in the past month. Patients were categorized as sleepy if at least 10 out of total 24 criteria were met.

### 2.4. Comorbidities and Medications

The study participants’ anthropometrics, smoking habits, medical history, and medications were obtained from the medical records as previously described [30]. Body mass index (BMI) was calculated by dividing weight (kg) by square of the height (m). BMI ≥ 30 kg/m^2^ was categorized as obesity.

### 2.5. MPO and MMP-9 Levels

All blood samples were collected in the morning (07:00–08:00 am) after overnight fasting. Ethylenediaminetetraacetic acid and serum tubes were used. The tubes were centrifuged, and the plasma/serum samples were aliquoted separately and stored at −70 °C. MPO and MMP-9 levels were studied in undiluted plasma samples with commercially available human cardiovascular disease biomarker multiplex assay kits (LINCOplexTM) according to the manufacturer’s instructions (Linco Research Inc.—St. Charles, MO, USA). The minimum detectable concentration (assay sensitivity) for MPO was 7 pg/mL, and for MMP-9 1 pg/mL. The concentration in all undiluted samples was within the standard curve for both biomarkers, ranging from 16 to 50.000 pg/mL for both MPO and MMP-9. The intra-assay and inter-assay variabilities (generated from the mean of the percentage coefficient of variability either from multiple reportable results across two different concentrations of the samples in one experiment or from two results each for two different concentrations of samples across several various experiments) were 4.5% to 12.3% and 8% to 16.3%, respectively.

### 2.6. Statistical Analyses

All statistical analyses were performed using SPSS^®^ 26.0 for Windows^®^ (SPSS Inc., Chicago, IL, USA). The normality assumptions for all variables in the current study were made with the Shapiro–Wilk test. Continuous variables were reported as mean with standard deviation (SD) or median with interquartile ranges (IQRs), and categorical variables were reported as percentages. All patients were dichotomized into high or low MPO and MMP-9 groups, based on the median cut-off values, and the baseline differences between the groups were analyzed with the Mann–Whitney U test for continuous variables and the Chi-square test for categorical variables. In addition, unadjusted and adjusted odd ratios (ORs) with 95% confidence intervals (CIs) for all variables associated with the high MPO and MMP-9 levels were performed, respectively. Age, sex, obesity, OSA, history of AMI, and current smoking at baseline were included in the multivariate models. All statistical tests were two-sided, and a *p*-value < 0.05 was considered significant.

## 3. Results

The mean age of the eligible patients (*n* = 502) was 63.9 (8.6) years, and 84% of the study participants were male. The median value of circulating MPO was 115.9 (69.4–183.5) ng/mL, and MMP-9 was 269.2 (196.4–364.1) ng/mL with no significant difference between CAD patients with OSA versus without OSA (for MPO 116.2 (70.3–188.2) vs. 113.2 (66.1–178.9) ng/mL, and for MMP-9 261.7 (188.9–366.3) vs. 290.3 (207.5–356.6) ng/mL, respectively). There were no significant differences between OSA patients with EDS versus without EDS (115.9 (75.4–194.0) vs. 117.6 (68.4–186.2) ng/mL for MPO, and 270.4 (188.5–387.1) vs. 261.6 (190.8–362.0) ng/mL for MMP-9, respectively).

As shown in Table 1, current smoking (*p* = 0.003), history of AMI (*p* = 0.007), and PCI at baseline (*p* = 0.014) as well as beta-blocker use (*p* = 0.013) were significantly more common among the participants with high MPO values compared with the patients with lower MPO levels.

For the circulating MMP-9 values, the participants with high MMP-9 levels were significantly younger, and the percentage of diuretic use was significantly lower whereas male sex, current smoking, history of AMI, PCI at baseline as well as calcium antagonist use were significantly more common among the participants of the high MMP-9 group compared to those with lower MPO levels (Table 2).

Within the non-smoker group, the median MPO values were similar between the never-smokers and former smokers (108 (63–176) vs. 109 (68–172) ng/mL) whereas the current smokers had significantly higher MPO levels (150 (97–214) ng/mL). Corresponding values for MMP-9 did not differ significantly between the never-smokers and former smokers (250 (174–328) vs. 260 (201–358) ng/mL) whereas the current smokers had significantly higher levels (308 (245–448) ng/mL).

As shown in Table 3, current smoking, AMI at baseline, and CABG at baseline were associated with high circulating MPO levels whereas OSA, AHI, ODI, and ESS were not. In the multivariate logistic regression analysis, current smoking (OR 1.73, 95% CI 1.06–2.84; *p* = 0.030) and beta blocker use (OR 1.81, 95% CI 1.04–3.16; *p* = 0.036) were significant determinants of high circulating MPO levels, independent of age, sex, obesity, OSA, history of AMI, and CABG at baseline (Figure 2).

As shown in Table 4, age, male sex, current smoking, AMI at baseline, CABG at baseline as well as diuretics and calcium channel blockers (CCB) use were associated with high circulating MMP-9 levels. In the multivariate logistic regression analysis, male sex (OR 2.07, 95% CI 1.23–3.50; *p* = 0.006) along with current smoking at baseline (OR 2.41, 95% CI 1.44–4.03; *p* < 0.001) and CCB use (OR 1.91, 95% CI 1.18–3.09; *p* = 0.008) remained significantly associated with high circulating MMP-9 levels, independent of age, obesity, OSA, diuretic use, history of AMI, and CABG at baseline (Figure 3). Neither OSA nor OSA severity in terms of AHI, ODI, lowest SpO2, and time spent below 90% SpO2 were associated with high MPO and high MMP-9 values in the entire cohort (Table 3 and Table 4); the results were similar in the sensitivity analysis after excluding patients with current smoking at baseline.

## 4. Discussion

The current analysis revealed that current smoking, but not OSA, was the significant determinant of high circulating MPO and MMP-9 levels in this revascularized CAD cohort. The degree of the OSA severity and coexisting daytime sleepiness were not associated either with the increased levels of these biomarkers.

Several experimental and clinical studies have shown that plasma MPO and MMP-9 concentrations are elevated in CAD patients [11,12,17,18,19]. As a marker of oxidative stress, MPO is higher after CAD events, which has been demonstrated to predict poor prognosis in CAD patients [13]. In a meta-analysis including 13 studies with 9090 subjects with acute coronary syndrome and a median follow-up of 11.4 months, Kolodziej and coworkers suggested that high MPO levels significantly predicted mortality with an OR of 2.0 (95% CI 1.4–2.9) [13]. Moreover, in a large population-based nested case–control study, the risk of future CAD increased in consecutive quartiles of MPO concentration, with an OR of 1.5 in the highest versus the lowest quartiles (95% CI 1.2–1.8) [14]. After adjustment for traditional risk factors, the OR in the highest quartile remained significant (OR 1.4 (95% CI 1.1–1.7) [14]. Similarly, MMP-9 is also linked to worse prognosis in patients with CAD [20]. Zhu and colleagues studied the association of plasma MMP-9 levels at admission with the in-hospital mortality in 155 patients who received the emergency percutaneous coronary intervention (PCI) following an AMI [20]. The mean plasma level of MMP-9 (528.9 ± 191.6 ng/mL) was significantly higher in patients who died (*n* = 24) than that in survivors (385.4 ± 236.0 ng/mL) during 14 days of hospitalization [20]. In a stepwise multiple logistic regression model, the MMP-9 level was associated with a five-fold risk increase for in-hospital death after adjustment for all other traditionally recognized risk factors [20].

Less is known regarding the association between OSA and MPO. In a small study consisting of 32 patients with OSA recruited from a sleep clinic, Akpınar and colleagues reported higher MPO values in saliva but not in serum compared to the values in 24 age- and gender-matched healthy controls, and there was a moderate positive correlation between salivary MPO levels and AHI [21]. In another study, Hanikoglu and colleagues found higher serum MPO levels in 59 patients with OSA compared with the values in 26 healthy controls, and there was a significant but weak correlation between MPO levels and AHI [22]. In a later study of 50 patients with OSA and 20 simple snorers, Arisoy and colleagues found no significant differences in MPO levels between the groups [23].

Literature regarding the association between OSA and MMP-9 is relatively better documented. Tazaki and colleagues reported higher serum levels and enzymatic activity of MMP-9 in 44 OSA patients compared with the values in 18 healthy controls, and one-month CPAP treatment successfully decreased both the level and activity of the MMP-9 [24]. Higher levels of MMP-9 were also reported in another study conducted among 51 male patients with OSA compared with 25 otherwise healthy obese men [26], and AHI was significantly correlated with MMP-9 levels after adjustment for age and BMI [26]. On the other hand, MMP-9 levels in children with OSA did not differ significantly from the age-matched healthy controls [25]. A recent meta-analysis, including 15 eligible studies, showed increased MMP-9 levels in OSA patients with a standardized mean difference (SMD) of 1.37 (95% CI 1.15–1.59) [28]. Furthermore, there was a significant dose–response relationship between the OSA severity in terms of AHI and MMP-9 levels [28]. Nevertheless, the small sample sizes of these case–control studies have been a major limitation [28].

Data regarding the MPO and MMP-9 values in patients with both CAD and OSA are scarce. In a case–control study including 209 OSA patients, circulating MMP-9 levels were significantly higher in patients with concomitant CAD [29]. On the other hand, the percentage of current smokers was reported to be two times higher among OSA patients with coexisting CAD than in those without CAD, which limits the real contribution of OSA per se to the increased circulating MMP-9 levels.

To our best knowledge, the current study is, to date, the largest study with 502 adults with CAD and concomitant OSA (*n* = 391) and no-OSA (*n* = 111) investigating the association of OSA with MPO and MMP-9 in CAD. The secondary analysis of a subgroup of patients from the ISAACC cohort included 361 adults with acute coronary syndrome and found no significant differences between patients with OSA versus no-OSA [30], which is in line with our results. In our RICCADSA cohort, the proportion of current smokers at baseline were 17% among the patients with OSA and 26% in no-OSA whereas corresponding numbers were 44.5% vs. 47.4%, respectively, in the ISAACC trial. Our results strongly indicate that current smoking is a major contributing factor to increased MPO and MMP-9 levels in patients with CAD regardless of OSA, which may partly explain the lack of significant differences in major adverse outcomes in OSA versus no-OSA in the ISAACC trial.

Current smoking seems to be a primary trigger for higher expressions of MMP-9 levels after CAD events [17,36]. In vitro studies showed that cigarette smoke increases the secretion of MMP-9 in cultured endothelial cells [37]. Similarly, a recent meta-analysis revealed that female sex and smoking status have a strong influence on the prognostic value of MPO in terms of mortality and recurrent MI (metaregression coefficient −8.616, 95% CI −14.59 to −2.63 and 4.88, 95% CI 0.76–9.01, respectively) [13].

The current study has several limitations. Firstly, the power analyses for the main RICCADSA trial were made for the primary outcome and not for the secondary outcomes addressed in this study. However, the sample size of our study is considerably large compared to previously published studies on this topic. Secondly, sleepiness is defined based on a threshold value in ESS that can be questionable in the CAD population [38]. However, this questionnaire is well documented and commonly used in clinical studies [35]. Lastly, our study population consists of revascularized CAD patients, and a control group without CAD is not included. Further longitudinal analysis of the natural course of MPO and MMP-9 levels in CAD with and without OSA, the impact of CPAP treatment on these biomarkers as well as the association of changes from baseline with the major adverse outcomes in the RICCADSA cohort are in progress.

## 5. Conclusions

Current smoking, but not OSA, was significantly associated with high MPO and MMP-9 levels in this revascularized CAD cohort. Smoking status should be seriously taken into consideration while evaluating the effects of OSA and its treatment on long-term adverse cardiovascular outcomes in adults with CAD.

## Figures and Tables

**Figure 1 jcm-12-04053-f001:**
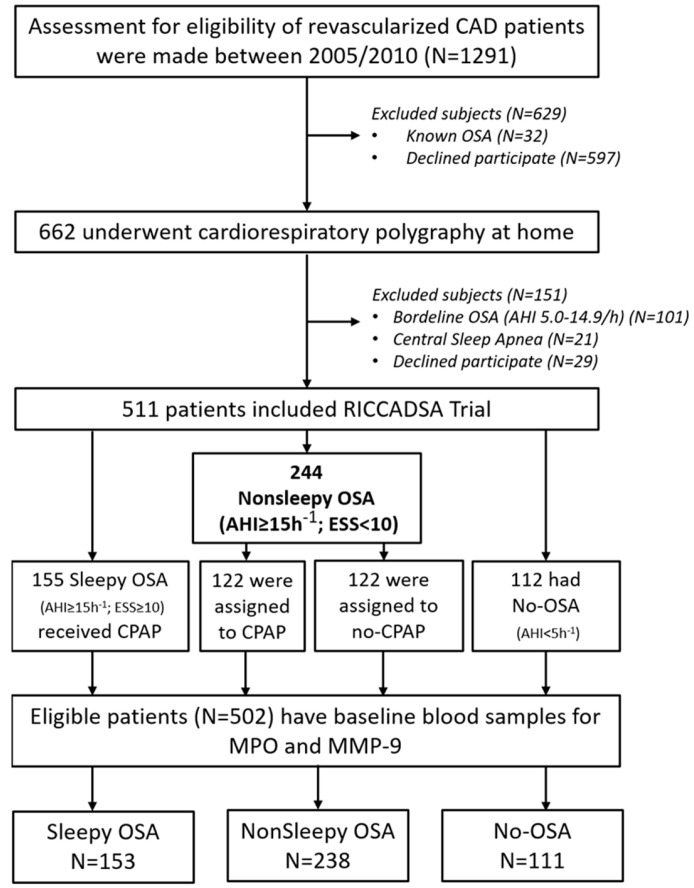
Flow-chart of the study participants. AHI, apnea–hypopnea index; CAD, coronary artery disease; MMP-9, matrix metalloproteinase-9; MPO, myeloperoxidase; OSA, obstructive sleep apnea; RICCADSA, randomized intervention with CPAP in coronary artery disease and sleep apnea.

**Figure 2 jcm-12-04053-f002:**
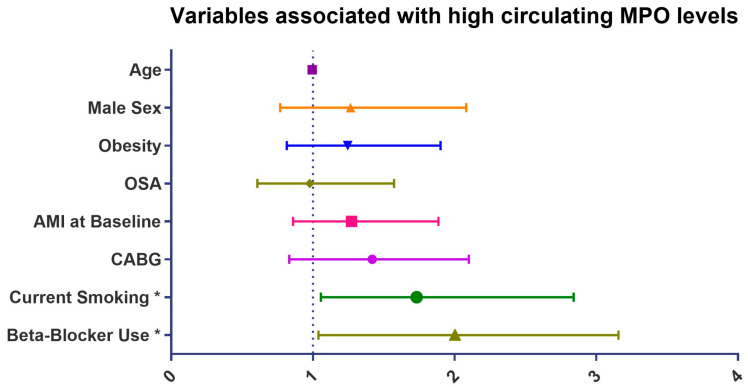
Variables associated with high circulating MPO levels in the multivariate logistic regression analysis. Abbreviations: AMI, acute myocardial infarction; CABG; Coronary artery bypass graft; MMP-9, myeloperoxidase-9; MPO, myeloperoxidase; OSA, obstructive sleep apnea. * *p* < 0.05.

**Figure 3 jcm-12-04053-f003:**
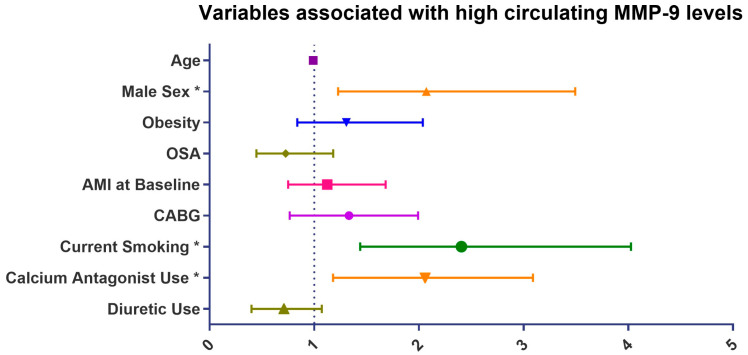
Variables associated with high circulating MMP-9 levels in the multivariate logistic regression analysis. Abbreviations: AMI, acute myocardial infarction; CABG; Coronary artery bypass graft; MMP-9, myeloperoxidase-9; MPO, myeloperoxidase; OSA, obstructive sleep apnea. * *p* < 0.05.

**Table 1 jcm-12-04053-t001:** Demographic and clinical characteristics of the patients dichotomized according to the median values of the circulating MPO levels.

	Low MPO Group	High MPO Group	
	MPO < 116 ng/mL (*n* = 252)	MPO ≥ 116 ng/mL (*n* = 250)	*p* Values
Age, years	64.9 (58.9–70.8)	63.5 (57.8–69.9)	0.187
Male sex, %	82.1	85.6	0.292
BMI, kg/m^2^	27.7 (25.5–30)	27.9 (25.6–30.8)	0.166
Obesity, %	24.2	30.0	0.144
Current smokers *, %	13.5	24.0	0.003
OSA, %	77.4	78.4	0.783
Sleepiness, %	33.3	30.8	0.543
AHI, events/h	20.8 (15.5–31.3)	21.6 (15.4–36.1)	0.823
ODI, events/h	10.1 (3.3–19.7)	10.5 (3.9–21.7)	0.607
ESS score	8.0 (4.0–10.0)	7.0 (4.8.0–10.0)	0.597
Hypertension, %	56.3	59.2	0.518
Diabetes Mellitus, %	24.6	19.2	0.143
Pulmonary Disease, %	9.9	8.0	0.451
History of A.F., %	17.9	15.6	0.498
Baseline AMI *, %	45.6	57.6	0.007
Baseline ACS, %	65.1	72.8	0.062
Intervention-PCI *, %	71.0	80.4	0.014
Former PCI/Bypass, %	20.2	18.4	0.602
Stroke, %	8.3	5.2	0.170
Medications			
Diuretics, %	22.6	19.4	0.372
Beta-blockers *, %	82.7	90.3	0.013
Acetylsalicylic acid, %	90.5	90.4	0.948
Warfarin, %	7.4	7.3	0.960
Clopidogrel, %	53.7	61.4	0.080
CAs, %	16.9	21.8	0.169
ACEIs, %	42.0	46.4	0.327
AT2As, %	14.8	13.3	0.631
LLDs, %	93.4	94.8	0.528

Definitions of abbreviations: ACEI, Angiotensin-converting enzyme inhibitors; ACS, acute coronary syndrome; AHI, apnea–hypopnea index; AMI, acute myocardial infarction; AT2A, angiotensin II antagonist; BMI, Body mass index; ESS, Epworth Sleepiness Scale; LLD, lipid-lowering drug; MPO, myeloperoxidase; ODI, oxygen desaturation index; PCI, percutaneous intervention. * *p* < 0.05.

**Table 2 jcm-12-04053-t002:** Demographic and clinical characteristics of the patients dichotomized according to the median values of the circulating MMP-9 levels.

	Low MMP-9 Group	High MMP-9 Group	
	MMP-9 < 269 ng/mL (*n* = 253)	MMP-9 ≥ 269 ng/mL (*n* = 249)	*p* Values
Age *, years	65.0 (59.2–71.1)	63.1 (58.0–69.0)	0.026
Male sex *, %	79.4	88.4	0.007
BMI, kg/m^2^	27.6 (25.5–30.1)	28.1 (25.6–30.5)	0.279
Obesity, %	25.7	28.5	0.477
Current smokers *, %	11.1	26.5	<0.001
OSA, %	80.2	75.5	0.201
Sleepiness, %	32.0	32.1	0.978
AHI, events/h	21.8 (15.9–32.8)	20.4 (15–32.3)	0.219
ODI, events/h	10.9 (4.1–20.7)	9.9 (3.7–20.4)	0.354
ESS score	7.0 (4.0–10.0)	7.0 (5.0–10.0)	0.676
Hypertension, %	55.7	59.8	0.351
Diabetes Mellitus, %	21.7	22.1	0.925
Pulmonary Disease, %	9.9	8.0	0.468
History of A.F., %	15.8	17.7	0.577
Baseline AMI *, %	47.0	56.2	0.039
Baseline ACS, %	66.4	71.5	0.219
Intervention-PCI *, %	71.5	79.9	0.029
Former PCI/Bypass, %	22.1	16.5	0.108
Stroke, %	6.7	6.9	0.942
Medications			
Diuretics *, %	25.1	16.8	0.024
Beta-blockers, %	84.6	88.5	0.204
Acetylsalicylic acid, %	90.3	90.6	0.901
Warfarin, %	6.9	7.8	0.691
Clopidogrel, %	54.4	60.8	0.149
CAs *, %	15.8	23.0	0.045
ACEIs, %	43.7	44.7	0.833
AT2As, %	13.4	14.8	0.657
LLDs, %	93.1	95.1	0.356

Definitions of abbreviations: ACEI, Angiotensin-converting enzyme inhibitors; ACS, acute coronary syndrome; AHI, apnea–hypopnea index; AMI, acute myocardial infarction; AT2A, angiotensin II antagonist; BMI, Body mass index; ESS, Epworth Sleepiness Scale; LLD, lipid-lowering drug; MMP-9, matrix metalloproteinase 9; ODI, oxygen desaturation index; PCI, percutaneous intervention. * *p* < 0.05.

**Table 3 jcm-12-04053-t003:** Unadjusted Odds Ratios (OR) with 95% confidence intervals (CI) for variables associated with high MPO levels in the RICCADSA cohort.

High MPO Levels	Bounds for 95% CI			
	OR	Lower	Upper	*p* Values
Age	0.99	0.97	1.01	0.166
Male sex	1.29	0.80	2.08	0.293
BMI	1.04	0.99	1.08	0.115
Obesity	1.34	0.90	1.99	0.145
Current smoking *	2.03	1.27	3.22	0.003
OSA	1.06	0.70	1.62	0.783
AHI	1.00	0.99	1.01	0.444
ODI	1.01	0.99	1.02	0.408
ESS score	0.99	0.94	1.03	0.490
Hypertension	1.12	0.79	1.60	0.518
Diabetes Mellitus	0.73	0.48	1.12	0.144
Pulmonary disease	0.79	0.43	1.46	0.452
History of atrial fıbrillation	0.85	0.53	1.36	0.498
AMI at baseline *	1.62	1.14	2.30	0.007
ACS at baseline	1.44	0.98	2.10	0.062
CABG at baseline *	1.67	1.11	2.53	0.015
Former revascularization	0.89	0.57	1.39	0.602
Stroke	0.61	0.30	1.24	0.173
Medications				
Diuretics	0.82	0.53	1.27	0.373
Beta-blockers *	1.95	1.14	3.34	0.015
Acetylsalicylic acid	0.98	0.54	1.79	0.948
Warfarin	0.98	0.50	1.94	0.960
Clopidogrel	1.38	0.96	1.97	0.080
Calcium antagonist	1.37	0.87	2.15	0.170
ACEI	1,20	0.84	1.71	0.327
AT2As	0.88	0.53	1.47	0.631
LLDs	1.27	0.60	2.71	0.529

Definitions of abbreviations: ACEI, Angiotensin-converting enzyme inhibitors; ACS, acute coronary syndrome; AHI, apnea–hypopnea index; AMI, acute myocardial infarction; AT2A, angiotensin II antagonist; BMI, Body mass index; CCB, calcium channel blocker; CABG; Coronary artery bypass graft; ESS, Epworth Sleepiness Scale; LLD, lipid-lowering drug; MPO, myeloperoxidase; ODI, oxygen desaturation index; PCI, percutaneous intervention. * *p* < 0.05.

**Table 4 jcm-12-04053-t004:** Unadjusted Odds Ratios (OR) with 95% confidence intervals (CI) for variables associated with high MMP-9 levels in the RICCADSA cohort.

High MMP-9 Levels	Bounds for 95% CI			
	OR	Lower	Upper	*p* Values
Age *	0.97	0.95	0.99	0.009
Male sex *	1.96	1.20	3.21	0.007
BMI	1.02	0.98	1.07	0.312
Obesity	1.15	0.78	1.71	0.477
Current smoking *	2.90	1.79	4.70	<0.001
OSA	0.76	0.50	1.16	0.202
AHI	1.00	0.99	1.01	0.361
ODI	1.00	0.98	1.01	0.504
ESS score	1.01	0.97	1.05	0.727
Hypertension	1.18	0.83	1.69	0.352
Diabetes Mellitus	1.02	0.67	1.56	0.925
Pulmonary disease	0.80	0.43	1.48	0.469
History of atrial fibrillation	1.14	0.72	1.83	0.577
AMI at baseline *	1.45	1.02	2.06	0.040
ACS at baseline	1.27	0.87	1.85	0.219
CABG at baseline *	1.58	1.05	2.39	0.029
Former revascularization	0.69	0.44	1.09	0.109
Stroke	1.03	0.51	2.06	0.942
Medications				
Diuretics *	0.60	0.39	0.94	0.025
Beta-blockers	1.40	0.83	2.37	0.206
Acetylsalicylic acid	1.04	0.57	1.90	0.901
Warfarin	1.15	0.58	2.26	0.692
Clopidogrel	1.30	0.91	1.86	0.149
Calcium antagonist *	1.59	1.01	2.50	0.046
ACEIs	1.04	0.73	1.48	0.833
AT2As	1.12	0.67	1.87	0.657
LLDs	1.43	0.67	3.06	0.358

Definitions of abbreviations: ACEI, Angiotensin-converting enzyme inhibitors; ACS, acute coronary syndrome; AHI, apnea–hypopnea index; AMI, acute myocardial infarction; AT2A, angiotensin II antagonist; BMI, Body mass index; CCB, calcium channel blocker; CABG; Coronary artery bypass graft; ESS, Epworth Sleepiness Scale; LLD, lipid-lowering drug; MMP-9, matrix metalloproteinase 9; ODI, oxygen desaturation index; PCI, percutaneous intervention. * *p* < 0.05.

## Data Availability

Individual participant data that underlie the results reported in this article can be obtained by contacting the principal investigator of the RICCADSA trial: yuksel.peker@lungall.gu.se.

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
