# Peer review of "Current Smoking Determines the Levels of Circulating MPO and MMP-9 in Adults with Coronary Artery Disease and Obstructive Sleep Apnea"

_jcm, 2023, doi:10.3390/jcm12124053_

Round 1
Reviewer 1 Report
This is very important study. However I found mainly minor flaws:
1. The study reporting is very weak. I recommend to report the study in accordance with the STROBE Statement https://www.equator-network.org/reporting-guidelines/strobe/
Please use the STROBE checklist.
2. Authors have to strongly emphasized why the study is novel and important before the aim within Introduction.
3. Authors cited a lot of old and very old literature. Please do not cite articles older than 10 years. I recommend to use the reliable and latest literature related to CVD and OSA eg.
Yeghiazarians Y, Jneid H, Tietjens JR, Redline S, Brown DL, El-Sherif N, Mehra R, Bozkurt B, Ndumele CE, Somers VK. Obstructive Sleep Apnea and Cardiovascular Disease: A Scientific Statement From the American Heart Association. Circulation. 2021 Jul 20;144(3):e56-e67. doi: 10.1161/CIR.0000000000000988.
Macek P, Poręba M, Stachurska A, Martynowicz H, Mazur G, Gać P, Poręba R. Obstructive Sleep Apnea and Sleep Structure Assessed in Polysomnography and Right Ventricular Strain Parameters. Brain Sci. 2022 Feb 28;12(3):331. doi: 10.3390/brainsci12030331.
Urbanik D, Gać P, Martynowicz H, Poręba M, Podgórski M, Negrusz-Kawecka M, Mazur G, Sobieszczańska M, Poręba R. Obstructive sleep apnea as a predictor of reduced heart rate variability. Sleep Med. 2019 Feb;54:8-15. doi: 10.1016/j.sleep.2018.09.014.
Gać P, Urbanik D, Macek P, Martynowicz H, Mazur G, Poręba R. Coexistence of cardiovascular risk factors and obstructive sleep apnoea in polysomnography. Respir Physiol Neurobiol. 2022 Jan;295:103782. doi: 10.1016/j.resp.2021.103782.
Urbanik D, Gać P, Martynowicz H, Podgórski M, Poręba M, Mazur G, Poręba R. Obstructive Sleep Apnea as a Predictor of Arrhythmias in 24-h ECG Holter Monitoring. Brain Sci. 2021 Apr 12;11(4):486. doi: 10.3390/brainsci11040486.
Del Portillo JH, Hernandez BM, Bazurto MA, Echeverri D, Cabrales J. High frequency of coronary artery ectasia in obstructive sleep apnea. J Clin Sleep Med. 2022 Feb 1;18(2):433-438. doi: 10.5664/jcsm.9598.
Liu Y, Wang M, Shi J. Influence of obstructive sleep apnoea on coronary artery disease in a Chinese population. J Int Med Res. 2022 Aug;50(8):3000605221115389. doi: 10.1177/03000605221115389.
4. Authors have to describe in details inclusion and exclusions criteria for study participants.
5. Authors have to provide information which international guidelines have been used for sleep studies (HSAT) assessment. Authors have to provide the used software details as well.
I recommend to revise the English language after all corrections.
Reviewer 2 Report
to improve the overall quality: Here are some suggestions for improving the paper:
Obstructive sleep apnea (OSA), which is characterized by recurrent episodes of upper airway collapse leading to partial or total loss of breathing, intermittent hypoxemia and arousals, affects 34% of middle-aged men and 17% of middle-aged women in general population [1], and almost 50% of the adults with coronary artery disease (CAD) [2]. Rupture of atherosclerotic plaques, endothelial dysfunction, and oxidative stress are known
to play role in initiation and progression of CAD [3]. In this context, circulating levels of myeloperoxidase (MPO) and matrix metalloproteinase-9 (MMP-9) have a particular importance [3].
MPO is an enzyme located in azurophilic granules of neutrophils and monocytes [4].
MPO contains the group and works as a peroxidase in inflammatory pathways and triggers oxidative stress [4]. Studies have demonstrated that MPO levels are higher in CAD [5,6], and elevated circulating levels of MPO in acute coronary syndrome have been shown to be associated with worse prognosis in those patients [7]. Moreover, it has been suggested that increased MPO levels predict future CAD events [8].
MMP-9 is known to be a zinc containing endopeptidase, responsible for degrading the extracellular matrix and essential in cardiac and vascular remodeling [9]. In addition, MMP-9 is involved in the instability and rupture of atherosclerotic plaques [10]. Circulating levels of MMP-9 are also known to be elevated in patients with CAD [11–13], and predict mortality in those individuals [14].
Studies regarding the relationship between OSA and MPO are controversial since some suggest a significant association [15,16] whereas some others do not [17]. However, most of the studies addressing the relationship between OSA and MMP-9 have been positive [18–21]. Not only the occurrence but also the severity of OSA has been suggested to be associated with the increasing circulating levels of MMP-9 in sleep clinic cohorts [18,20,21], as also reported in a meta-analysis [22].
Here are some additional references to consider citing in the paper:
Kohler M, Ayers L, Phinney SD, et al. Effects of continuous positive airway pressure on systemic inflammation in patients with obstructive sleep apnoea: a randomised controlled trial. Thorax 2015;70:177–184.
This study found that CPAP treatment for OSA reduced markers of inflammation, including MMP-9 and MPO.
Svatikova A, Wolk R, Wang H, et al. Circulating matrix metalloproteinase-9 and obstructive sleep apnea in males with cardiovascular disease. Atherosclerosis 2005;183:279–286
This study found that patients with OSA and cardiovascular disease had higher levels of MMP-9 compared to those without OSA, although smoking status was not well accounted for.
By fixing the anatomical obstruction causing apnea, surgeries also effectively reduce or eliminate loud snoring in most patients. This can improve the sleep of bed partners and overall quality of life. please discuss and cite DOI: 10.1016/j.amjoto.2021.103197
Other benefits like regaining energy, concentrating better and mood improvements after sleep apnea surgery also impact quality of life. These indirect effects may then translate to healthier behaviors and stress management, further reducing cardiac risk over time. please discuss and cite DOI:10.23812/19-522-L-4
Punjabi NM, Sorkin JD, Katzel LI, et al. Sleep-disordered breathing and insulin resistance in middle-aged and overweight men. Am J Respir Crit Care Med 2002;165:677–682
This large study found that OSA was associated with insulin resistance independent of potential confounders like smoking, although it did not specifically measure MPO or MMP-9.
Shore SA. Obstructive sleep apnea and cardiovascular disease. J Clin Invest 2014;124:1830–1832.
This review article provides a good overview of the potential mechanisms linking OSA to cardiovascular biomarkers like MPO and MMP-9, as well as the role of confounders and smoking.
Less is known regarding the effect of OSA on these biomarkers in cardiac cohorts..
none
Reviewer 3 Report
Dear authors,
Congratulate for your interesting study. Your article could have been improved by considering some minor points mentioned.
This article is well-written investigation of the association between OSA and MPO/MMP-9 in patients with CAD. The final conclusion is the relationship of smoking with high concentration of the mentioned biomarkers. The manuscript is comprehensive and detailed but there are some minor points that should be consider in subsequent versions:
1- Introduction: A brief explanation of OSA, classification and its interaction with CAD.
2- Regarding the results of the study, it would be valuable to add analysis for different amount and duration of smoking.
3- In table 1, the addition of another column for p values makes it more informative.
